# Caregivers’ and Health Extension Workers’ Perceptions and Experiences of Outreach Management of Childhood Illnesses in Ethiopia: A Qualitative Study

**DOI:** 10.3390/ijerph18073816

**Published:** 2021-04-06

**Authors:** Atkure Defar, Kassahun Alemu, Zemene Tigabu, Lars Åke Persson, Yemisrach B. Okwaraji

**Affiliations:** 1Ethiopian Public Health Institute, Addis Ababa P.O. Box 1242, Ethiopia; Lars.Persson@lshtm.ac.uk (L.Å.P.); yemisrach.okwaraji@lshtm.ac.uk (Y.B.O.); 2Department of Epidemiology and Biostatistics, Institute of Public Health, College of Medicine and Health Sciences, University of Gondar, Gondar P.O. Box 196, Ethiopia; kassalemu@gmail.com; 3Department of Paediatrics and Child Health, School of Medicine, College of Medicine and Health Sciences, University of Gondar, Gondar P.O. Box 196, Ethiopia; zemene.tigabu@gmail.com; 4Department of Disease Control, London School of Hygiene and Tropical Medicine, London WC1E 7HT, UK

**Keywords:** outreach services, home-based services, sick child, focus group discussion, key informant interview, Ethiopia

## Abstract

Introduction: Ethiopian Health Extension Workers provide facility-based and outreach services, including home visits to manage sick children, aiming to increase equity in service coverage. Little is known about the scope of the outreach services and caregivers’ and health workers’ perceptions of these services. We aimed at exploring mothers’ and health extension workers’ perceptions and experiences of the outreach services provided for the management of childhood illnesses. Methods: Four focus groups and eight key informant interviews were conducted. A total of 45 community members participated. Interviews were recorded, transcribed verbatim, and translated into English. We applied thematic content analysis, identified challenges in providing outreach services, and suggestions for improvement. We balanced the data collection by selecting half of the participants for interview and focus group discussions from remote areas and the other half from areas closer to the health posts. Results: Mothers reported that health extension workers visited their homes for preventive services but not for managing childhood illnesses. They showed lack of trust in the health workers’ ability to treat children at home. The health extension workers reported that they provide sick children treatment during outreach services but also stated that in most cases, mothers visit the health posts when their child is sick. On the other hand, mothers considered distance from home to health post not to be a problem if the quality of services improved. Workload, long distances, and lack of incentives were perceived as demotivating factors for outreach services. The health workers called for support, incentives, and capacity development activities. Conclusions: Mothers and health extension workers had partly divergent perceptions of whether outreach curative services for children were available. Mothers wanted improvements in the quality of services while health workers requested capacity development and more support for providing effective community-based child health services.

## 1. Introduction

The Sustainable Development Goals aim at ensuring that all people get access to the health services they need without facing financial deprivation [1]. Thus, health systems need to be strengthened to reach a universal health coverage and end preventable maternal and child deaths. Programs need to reach every community and household to provide evidence–based interventions—also in remote communities [2].

Community health programs have the potential of ensuring equity in coverage of essential maternal and child health services [3]. However, women and children from socioeconomically and geographically disadvantaged households use these services less than those in favorable settings [4,5]. The reported barriers to utilization of primary care services for sick children are multiple and vary between countries and contexts. It might be a lack of knowledge on what the first-line healthcare services can offer, shortage of health facilities within accessible distance, missing skills or trust in health workers, stock-out of medicines and supplies, irregular facility opening hours and low quality of services at the facility [5,6]. Home-based and other outreach curative services by community health workers might be one of the strategies to overcome these gaps [7,8].

Various strategies were employed by the global community to scale-up effective child health interventions [9,10]. The integrated Community Case Management is effective in reducing under-five mortality if implemented with sufficient coverage [9]. Community health workers are able to manage common childhood illnesses outside health facilities, which include identifying and treating sick children at their homes. This approach is supported by the WHO and UNICEF to enhance survival of newborns and children as part of the efforts to reach the Sustainable Development Goals by 2030 [11,12].

To improve access to primary healthcare services for children, Ethiopia implemented community-based interventions through the Health Extension Program, since 2003 [13,14]. These efforts faced several implementation challenges [15]. The program was designed and implemented to empower households and communities to promote health and prevent diseases [11]. The strategies included home-based preventive and curative child health services to enhance equity in coverage [16]. Various contextual factors limited the scale-up of these outreach activities [17]. Services provided in outreach activities had greater equity in coverage, although these were highly affected by geographical characteristics [18,19]. Thus, in this qualitative study, we aimed at exploring mothers’ and health extension workers’ perceptions and experiences of the outreach services provided for the management of childhood illnesses.

## 2. Methods

### 2.1. Study Design and Study Areas

A qualitative study was conducted in October 2019, including Focus Group Discussions (FGD) with mothers of under-five children and key informant interviews with health extension workers. The study was conducted in two districts, Dangla and Ankesha, of the Awi zone in the Amhara regional state, Ethiopia. Two easily accessible and two remote kebeles (the lowest administrative unit) were purposively selected for the discussions and the interviews. This selection was done to include variation in geographic and distance locations when discussing home-based care.

### 2.2. Sampling Procedure and Recruitment of the Study Participants

The research team had communicated with the regional health bureau and the district health offices of the study area, to inform and get administrative approval for the study. We identified one person, who was working within the local health system and knew the setting well, and received assistance in the selection of discussants and key informants. We introduced the project aim and explained the criteria for participant selection.

In total, four FGDs with mothers or other caregivers of under-five children and eight key informant interviews with health extension workers were included. Two FGDs were conducted in the easily accessible area and two in the remote kebeles (the lowest local administrative unit). Four health extension workers from each district were interviewed. All participants were purposively selected.

Mobilizers contacted mothers with under-five children, who were available at the time of recruitment in the study areas, to invite to participate. An attempt was made to select participants from different localities within the rural area—closer and more distant from the health post. The inclusion criteria for participation in FGDs were residence in the study setting and being the primary caregiver for at least one child under the age of five years.

We identified one health post in an easily accessible area from the main road and one remotely located. After this, we obtained the information on health extension workers at these facilities and selected one randomly at each health post. The interviews of the health extension workers were done at their health posts.

The selection of both mothers (as service users) and health extension workers (care providers) was done to obtain different perspectives on outreach healthcare services for children. Inclusion of additional groups and interviews continued until saturation was reached, i.e., when additions did not yield new information [20]. Thus, two FGDs from each site were conducted. Each FGD involved 7 to 9 participants.

### 2.3. Data Collection

Data were collected using a pre-tested, semi-structured guide developed by the authors (Appendix A). Data were collected by two trained female interviewers with Master of Public Health qualifications. We used female data collectors since all respondents were women. We benefitted from a local mobilizer to identify study participants, the place for interview and other logistics. A facilitator managed the FGDs and created a comfortable environment for participants to openly voice their opinions. We also used a translator for one of the FGDs in the Ankesha district.

The FGDs with mothers of under-five children included their perceptions and experiences of community-based interventions, and facilitators and barriers for home-based care. They were also asked about the organization of the outreach efforts and possible improvement strategies.

A total of eight health extension workers were interviewed about their experiences and perspectives on outreach service provision (using a backpack, foldable register, chart booklet, basic medicines, and supplies), community perception, and acceptability of home-based care, including detailed information on the services they provided. The interview with the HEW was finalized without any problem.

### 2.4. Data Management and Analysis

Before beginning the data collection and coding process, pre-set codes were derived from the research objectives and questions (A.D. and Y.O.). These were—knowledge and experience of the community about the outreach services, the experience of the health extension workers of outreach service provision, the barriers and enablers of service provision, and suggested improvements. Additionally, ideas, concepts, actions, relationships, meanings, etc., that appeared in the data and were different from the pre-set codes were used as emerging codes. The tape-recorded discussions and interviews were transcribed verbatim and translated into English. Field notes were translated into English. Data were analyzed and compiled using a thematic approach, with an ongoing content analysis [21]. This consisted of multiple readings of transcripts from group discussions and interviews to ensure familiarity and understanding of the data, to identify recurring ideas, and capture these in reflective notes.

Data from interviews and focus groups were coded within broad themes, using an Excel spreadsheet [22]. Coding was based on reading each transcript and identifying the underlying meaning of each segment of text and was done by A.D. For each segment, considerations were made on the content of the segment, what message it gave, what stuck out, and how it differed from or repeated other segments. Each segment of text was given one or more codes reflecting its underlying meaning. Codes of similar concepts were sorted and merged into themes. Themes and codes were refined and adjusted by looking for patterns, links, and contradictions within themes. Emerging themes were developed from the expanded interviews and discussions (A hybrid coding approach). A subset of transcripts was independently coded by Y.O., who also reviewed A.D.’s coding regularly. Discrepancies in coding were resolved through coding meetings, which were held several times during the analysis period. Selected quotations were presented in the text. Data credibility was checked by triangulating data between the respondent groups and between data collection methods. After repeated reading of the transcripts, coding frames were generated.

## 3. Results

### 3.1. Characteristics of the Participants

All participants were women, which was consistent with the socio-cultural norm in the study context where the primary caregivers of children were the mothers or other female caregivers. All 37 participants in the FGDs had at least one child under the age of five years. The mean age of FGD participants was 28 years. The eight health extension workers had in most cases two years of schooling. Their work experience ranged from 4 to 17 years.

### 3.2. Themes Formation

We identified two major themes. The first was mothers’ perceptions and experiences of outreach services with two sub-themes—types of outreach services and treating sick children at home. The second was the health extension workers’ perceptions of outreach services. In this theme, there were three interlinked subthemes—types of outreach services provided, challenges in providing outreach services, and activities needed to improve these services.

#### 3.2.1. Mothers’ Experiences and Perceptions of Outreach Services

##### Outreach Services Received

The mothers stated that sanitation, nutrition, and immunization were the most common outreach services provided. These services included among other things breast-feeding and infant feeding counselling, advice on separating animals and humans in living areas, building toilets, hand washing, and using bed nets. This was resonated by:
“*The health extension worker teaches us based on the gaps we have. She teaches about how to feed our children. She tells us about antenatal care follow up, and if there is a problem that is beyond her capacity, she refers to the health centre. We take our children to the health post when we have a problem and if it is beyond her capacity then she refers us to the health centre…*” (Mother, 26 years, FGD).

The reason for health extension workers to make house-to-house visits was not primarily to identify and treat sick children. Rather, the intention was health promotion and preventive activities, which the mothers perceived as the health extension workers’ first priority. This idea was echoed by:

“*The main reason for the house visits was to check that we are using a bed net. And to check our toilet and to make sure that we have a separate house for the animals…*”
(Mother, 34 years, FGD).
The health extension worker visits our houses and tell or teach us to keep our house organized, to clean our toilet, and to have proper waste disposal and about child feeding (how to prepare porridge)(Mother, 38 years, FGD).

##### Treating Sick Children at Home

All mothers, regardless of distance to the health post, disapproved of health extension workers treating sick children at home. This idea was expressed by:
“*To be honest, we do not get treatment at home or community level, there is no such a thing as calling her [the health extension worker] and getting the treatment at the house*”. (Mother, 31 years, FGD).

Participants expressed concern about the health extension workers’ ability to examine a sick child without all necessary equipment that was only available at the health facility, such as laboratory tests, weighing scales, and medicines. In the words of one mother:

“*How can she give medicine without evaluating and diagnosing the disease, what does she use to evaluate and diagnose the disease at the household level?*”
(Mother, 43 years, FGD).

Another caregiver described the potential problem that could arise when the health extension worker tried to examine a sick child at home:

“*They can’t diagnose a disease with their naked eye. Let us say, my child got sick and how will I give him a medicine that they order at my house. You can only diagnose a disease by using different tests. If the child has fever and cough, he needs to be tested before prescribing a medicine…*”
(Mother, 34 years, FGD).

Even those who lived in a remote area preferred to go to the health post when their children were sick, so that they could be referred to the health center if the health extension worker was unable to treat the child.

“*It doesn’t matter whether we live far away or near the health post, we should still manage to come to the health post*”
(Mother, 29 years, FGD).

For many caregivers, the lack of trust in the health extension workers’ skills and the lack of diagnostic tools and essential drugs during home visits appeared to be an important consideration.

“*I don’t trust their knowledge that much*”
(Mother, 34 years, FGD).

“*…the health post should be well equipped for us to fully trust the medicines they prescribe. We wouldn’t go to the hospital if the health post was well equipped….*”
(Mother, 41 years, FGD).

Even if mothers gave their approval to the health extension workers treating their children at home, they had no means of communicating with the health workers when their children got sick—they did not own a landline or a mobile phone.

“*We don’t have phone and we can’t call them [the health extension workers] so we take the child to the health post instead*”
(Mother, 43 years, FGD).

#### 3.2.2. Health Extension Workers’ Experiences and Perceptions of Outreach Services

##### Outreach Services Provided and Its Challenges

The health extension workers reported that they provided services with the assumption that half of their time was spent in the community. One health extension worker was at the heath post while the other provided outreach services. This approach had increased care utilization at the health post and at home. A health extension worker stated:

“*The health post will not be closed at all times. If I am at the health post then my co-worker will be out in the field and vice versa… During the weekend (Sunday), we do awareness creation*”
(Health extension worker, interview).

The community did not seem to accept home-based curative services for sick children, but the health extension workers reported that they provided such outreach services. They stated:

“*When we go house-to-house [for outreach services] … we bring everything necessary, including vitamin A, deworming, MUAC, and for children we take zinc, ORS, amoxicillin. If they are experiencing diarrhoea, we treat them with ORS and zinc…*”
(Health extension worker, interview).
We have separate registration for house-to-house services and we have a chart booklet. We used to take rapid diagnostic test and
coartem (artemether-lumefantrine),
amoxicillin and so on. So, we assess if a child has diarrhoea or vomiting, then we diagnose. We treat diarrhoea with zinc and ORS and we treat pneumonia with amoxicillin.(Health extension worker, interview).

The health extension workers also reported that they provided outreach immunization service routinely:

“*…In addition, we [health extension workers] also provide immunization for children as an outreach activity and at the health post. We have three outreach programs per month, so we provide immunization according to this schedule*”
(Health extension worker, interview).

The common challenges experienced in providing outreach services were distance, lack of supplies, and lack of health extension worker commitment. Home-based care for sick children was uncommon in nearby households, since the caregivers usually brought their sick children to the health facility. The health extension workers also described how challenging and exhaustive it could be to reach some of the villages located in remote areas. A health extension worker presented this as follows:

“*There are some kebeles far from the health post and it might take us more than 30 min to get to them. And by the time we get there, we might already be tired and exhausted…*”
(Health extension worker, interview).

Some informants emphasized barriers such as distance would not be a problem if the health extension workers were more committed and motivated to do this job:

“*It can benefit all, but the question is whether we can provide this equally or not. We can treat sick children regardless of distance, it does not matter whether they are located near or far, if we come across them [sick children] we can still provide the treatment at home*”
(Health extension worker, interview).

The health extension workers reported that supplies did not limit them from providing outreach services. Mothers, on the other hand, wanted health posts to be better equipped to provide quality curative services and outreach services.

“*It has been a while since the integrated Community Case Management started but it was not strengthened, and we did not have all the necessary supply. But we were given a bag saying newborn as soon as this program was started, but ever since it was strengthened, we were given registration books to be used in the field…*”
(Health extension worker, interview).

##### Activities Needed to Improve Outreach Services

The key informants provided suggestions to improvements of services in their localities and community-based outreach programs. Review meetings, upgrading the service provision at the health post, and enhancing knowledge and skills of the health extension workers were some of the suggestions.

“*…I think it would be better to hire nurses at the health post to provide quality services and it will help us to gain experience from them as well*”
(Health extension worker, interview).

Performance review meetings that occurred quarterly helped the health extension workers to improve their work and technical skills. This appeared to strengthen outreach curative child health services.

“*Even though there is an interruption currently, the three-months review meeting was good because it helped us to know our strengths and weaknesses so it would be good to do that regularly*”
(Health extension worker, interview).

Awareness-creating strategies were discussed by the key informants. Churches, the 1 to 5 women’s development groups, influential people, and the kebele administrators were mentioned to reach the community. Such efforts could improve community awareness about the availability of services provided at the health facility and in the caregivers’ home.

“*The community is aware about the services through women’s development army, 1 to 5 structure, and influential people. In addition, during the implementation of community-based projects, we gathered 150 people in the kebele and we [the health extension workers] created awareness so they [the community] know that we provide house-to-house treatment services …*”
(Health extension worker, interview).

Other suggestions for improvement included career development, provision of residential houses near the health post, and increasing the health extension workers’ salary.

“*Motivation (salary and incentives) and further education…*”
(Health extension worker, interview).

“*If we have a house near the health post, we will not be tired and we can provide the home-based care better…*”
(Health extension worker, interview).

## 4. Discussion

Mothers reported that outreach services provided by the health extension workers’ to be limited in preventive and promotive services, such as advocacy for hygiene and nutrition and maternal health care, but also for immunization. With regards to treatment of sick children, the mothers questioned the ability of the health worker to provide such services at home. Health extension workers, on the other hand, stated that they provide home-based curative services, as they carry with them basic equipment and medicines during outreach services. To bridge such gaps in perception and understanding of services in terms of barriers to outreach services, health extension workers mentioned workload, long distances, lack of support, and incentives as factors that could demotivate to provide outreach services. They suggested health posts to be staffed with nurses to improve the quality of the curative services. Furthermore, they acknowledged the benefit of having non-governmental partners supporting the outreach services by providing them the necessary job aids and supplies, such as the chart booklet and drugs, and the provision of supportive supervision and clinical mentoring.

A low quality of primary child health services was one of the perceived barriers to the use of the health extension workers’ services. With ambitions to reach universal health coverage, as part of the sustainable development goals, the quality of health services is emphasized. In health systems with major quality problems, the most vulnerable segments of the population suffer more, and inequity in health increases [10]. In studies partly performed in the study areas of this qualitative study, we could show that health extension workers did not adhere to the integrated community case management guidelines for sick children, with negative consequences for diagnosis and treatment of children with suspected pneumonia, diarrhea, fever, and malnutrition [23,24]. In such situations, communities might have low trust in the curative services. Participatory community engagement and quality improvement strategies for mother and child healthcare might bridge such gaps in the perceptions and understanding of services, and improve the quality of the services provided in the community and at the health facility [25].

Mothers reported that distance to the health post was not a problem when deciding to seek care for the sick child or not. In a study in four Ethiopian regions, including the current study area, we found that distance and difference in altitude were not associated with sick child care seeking [26]. Other studies in Ethiopia and Tanzania reported that geographic access was a barrier to the utilization of health facilities [27,28] A study that included Kenya, Nigeria, and Niger revealed variation by country, where in Kenya, distance to the health facility was a challenge. In Nigeria, the influence of geography was limited as well as in Niger, where mothers, who lived in villages with a health post did not consider distance or location as a barrier to care-seeking for children [29]. The unfavorable topography with difficult terrain and rural remote populations with few households in each village make the outreach services difficult. However, in areas where community health workers or women’s groups provide home visits, service coverage of maternal and child health services might be enhanced [3]. Therefore, strategies need to be contextualized for outreach services.

In an evaluation of a community-based intervention in four Ethiopian regions, we showed that health posts with health extension workers were poorly equipped to provide services to sick children [30]. Therefore, equipping the health extension workers with skills, as well as essential medicines and other basic items is a prerequisite for improved quality services. A study conducted in Rwanda showed that the number of children receiving community-based treatment for diarrhea and pneumonia increased significantly one year after the implementation of the integrated community case management [31]. Similarly, a study conducted in Ethiopia revealed that performance review and clinical mentoring improved case management [32]. Support to the health extension workers through review meetings [33], training, and supplies (chart booklet, register, drug, etc.) [29] and other quality improvement strategies might improve quality, trust, and use of community-based child curative services [32,34].

The health extension workers identified several issues that could enhance their performance, such as support, incentives, lowering the workload, and capacity development activities. Similarly, another Ethiopian study revealed that the health extension workers’ motivation could be enhanced through regular review meetings and opportunities for career progression [33]. Similar to this, another study conducted to understand the relationships between HEWs and the community and health sector revealed that health extension workers’ relationships with the community and health sector could be affected as a result of inadequate support systems, lack of trust, communication, and expectations. Clearly defined roles at all levels and standardized support, monitoring, and accountability system, regular supervision, and training, increased their performance and maximized the their value and trust in the community and the health sector [35].

Community participation might contribute to the use of healthcare and increase the sense of ownership of the local health services. A systematic review showed that the Ethiopian women’s development groups contributed to enhanced maternal and child health service utilization [36]. However, in a recent study, the leaders of these groups lacked knowledge on maternal and child health, and were not so active regarding neonatal and child health issues [37]. Ideally, such community engagement groups might be a mechanism to share information from the health services and favorably influence the parents’ care-seeking behavior [38]. Positive development is possible when communities participate in problem identification and resolution, and are engaged as partners in improving child health [29].

## 5. Strengths and Limitations

This study included different perspectives—from participants living in remote as well as easily accessible communities, and information collected from service providers and users. The sample of informants for this qualitative study included different groups, representing varying relevant interests and perspectives. The sample was based on the concept of saturation; the inclusion continued until we reached saturation of information in the data collection.

There were some challenges during data collection. A few participants did not speak Amharic. We were able to manage this problem by involving a translator who spoke Amharic and the local language (Agew). Limitation of transferability of our study findings is a potential limitation. However, we believe that most results are relevant and highly likely to be transferable to similar settings in rural Ethiopia and beyond.

## 6. Conclusions

We explored mothers’ and health extension workers’ perceptions and experiences of the outreach services provided for the management of common childhood illnesses. Mothers viewed the health extension workers’ outreach services to be limited to preventive services, including vaccinations, and awareness creation activities. However, the health extension workers expressed their readiness to also provide curative services, but acknowledged that workload, long distances, and lack of incentives could demotivate outreach services.

In the Ethiopian context, further steps need to be taken ensuring outreach services, including curative child health services to remote communities and disadvantaged groups. Enhanced community ownership and partnership in the primary healthcare services might be a way forward. Motivational support to health extension workers and advocacy activities at the community might be needed to enhance community ownership of improved home-based curative services.

## Data Availability

The data that underpin this research is qualitative in nature. The data and interview transcripts cannot be made openly available due to reasons of confidentiality.

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
