# Peer review of "Caregivers’ and Health Extension Workers’ Perceptions and Experiences of Outreach Management of Childhood Illnesses in Ethiopia: A Qualitative Study"

_ijerph, 2021, doi:10.3390/ijerph18073816_

Round 1

Reviewer 1 Report

The manuscript presented is very interesting and highly relevant. It is worth paying attention to how health workers deal with childhood illnesses.

The paper is also very well written. 

Here are some recommendations for improvement of the submitted manuscript:
- It would be interesting if the authors could indicate bibliographical references to support the chosen study methodology.
- What exclusion and inclusion criteria were used for each group of participants?
- The authors seem to describe more the methodology followed to capture the information from the mothers' group and less the workers' group. It would be interesting to specify more the procedure with the latter.
- On the other hand, it would be relevant for the authors to include possible lines for the future based on the work carried out.

Reviewer 2 Report

Dear Authors,

Thank you for the opportunity to review your manuscript.  I offer the following recommendations for your consideration:

  1. Did this study receive Institutional Review Board approval?  If so, state the approval and institution granting the approval.
  2. The manuscript is quite repetitive throughout.  Revise the manuscript to reduce redundancy with respect to the findings.
  3. Please include the survey and interview instruments as an Appendix for those interested in replicating the work.
  4. Who was involved in the content thematic analysis? Describe this process in more detail.
  5. Discuss the findings in light of the small sample size.
  6. How do these findings impact policy and practice recommendations?  
  7. The discussion section should discuss the findings in light of the peer-reviewed literature.  Expand this discussion.

Round 2

Reviewer 1 Report

The modifications made by the authors comply with the requirements made by this author. 

Reviewer 2 Report

Thank you for addressing my comment.